# ENHANCING DECISION TREE LEARNING WITH DEEP NETWORKS

## ABSTRACT

Conventional approaches to (oblique) decision tree construction for classification are greedy in nature. They can fail spectacularly when the true labeling function corresponds to a decision tree whose root node is uncorrelated with the labels (e.g. if the label function is the product of the sign of a collection of linear functions of the input). We define a new figure of merit to capture the usefulness of a linear function/hyperplane in a decision tree that is applicable even in scenarios where greedy procedures fail. We devise a novel deep neural network architecture that is very effective at seeking out hyperplanes/half-spaces/features that score highly on this metric. We exploit this property in a subroutine for a new decision tree construction algorithm. The proposed algorithm outperforms all other decision tree construction procedures, especially in situations where the hyper-planes corresponding to the top levels of the true decision tree are not useful features by themselves for classification but are essential for getting to full accuracy. The properties of the deep architecture that we exploit to construct the decision tree are also of independent interest, as they reveal the inner workings of the feature learning mechanism at play in deep neural networks. [1]

## 1 INTRODUCTION

Neural networks and deep learning have demonstrated exceptional performance across diverse domains, but they raise at least as many questions as they give answers. Some of them being:

- What are the component parts and features of a neural network model, and how are they learned?

- What is the effect of different components on the final prediction function?

These and many more questions are mostly open and are a topic of current research. This leads us to believe we are in the heady days analogous to the time of the steam engine, before the discovery of thermodynamics. We believe making progress on these questions can supercharge the already tremendous impact of AI and ML even further.

Decision trees are a classic machine learning paradigm known for its simple, effective and interpretable models. While interesting in their own right, they form the ideal comparison point for making progress on the fundamental questions in deep learning. Several of the questions asked in the neural networks/deep learning setting, for which we have no answers, have elegant answers in the decision tree setting.

In this paper, we make observations based on some experimental findings that can aid in transferring some answers in the decision tree paradigm to the neural network paradigm. This paper can technically be just viewed as the design of a decision tree learning algorithm that exploits a property of deep networks to get better decision trees. However, we believe that the real impact of this paper lies in providing an alternative lens to study the key property of *feature learning* in neural networks.

---

[1] Our implementation and datasets are accessible at `https://github.com/anonymousgithub09/ICLR2024dlgndt.git`

## 1.1 Related Work

Decision trees and neural networks are some of the oldest paradigms of machine learning, and there have been muliple works using one in the aid of the other. Guo & Gelfand (1992) propose employing small multilayer nets at decision nodes to extract nonlinear features. Boz (2002) introduce DecText, a method effective in deriving high-fidelity trees from trained networks.Krishnan et al. (1999) employ a genetic algorithm to extract prototypes from trained networks, offering an alternative approach to traditional decision tree extraction. Schmitz et al. (1999) explore ANN-DT, a method for extracting rules from neural networks without assuming internal structures or data features, utilizing a novel attribute selection criterion.Sato & Tsukimoto (2001) introduced a method for extracting rules from neural networks using decision tree induction, providing a means of interpreting the network's behavior. Zhang et al. (2019) proposed a technique that employs decision trees to elucidate the specific reasons behind CNN predictions, breaking down feature representations into elementary object concepts for enhanced interpretability.

## 1.2 Contributions

We make 5 distinct contributions in this paper.

1. We identify a family of class label functions that can be efficiently represented by an oblique decision tree (ODT), but is immensely hard for any decision tree algorithm to learn from finite data.

2. We define a new quantity that measures the suitability of an ODT split criterion that is agnostic to how the other nodes (its ancestors particularly) in the tree are defined

3. We construct a novel deep architecture (DLGN) that outperforms kernel methods and is competitive with ReLU networks on classification tasks. It has a notion of features that corresponds to hyperplanes in the input space.

4. We show that these DLGN features exhibit strong tendencies to move towards the hyperplanes corresponding to the true ODT label function (if such a function exists)

5. We design a novel algorithm that exploits the above tendency to recursively construct a decision tree. This decision tree learning algorithm performs well even for label functions identified in the first point above, and is competitive with other algorithms in general.

## 1.3 Notation/Setup

In this paper, we consider a binary classification task, with training set $S = \{(\mathbf{x}_1, y_1), \ldots, (\mathbf{x}_n, y_n)\}$ where $\mathbf{x}_i \in \mathbb{R}^d$ is drawn from some distribution $D$, and $y_i = f^*(\mathbf{x}_i) \in \{+1, -1\}$. The ultimate goal is to generalize well by finding a classifier $f : \mathbb{R}^d \rightarrow \{+1, -1\}$ such that $f(\mathbf{x}) = f^*(\mathbf{x})$ with high probability over $\mathbf{x}$ drawn from D. We assume that $f^*$ can be represented efficiently by an oblique decision tree. For any positive integer $a$ we denote the set $\{1, 2, \ldots, a\}$ as $[a]$. We denote by $\mathbf{1}$(condition) as $0, 1$ valued variable that takes 1 if the condition is true and 0 otherwise.

## 2 Oblique Decision Trees

An oblique decision tree (ODT) (Bertsimas & Dunn, 2017; Murthy et al.; 1994; Wickramarachchi et al., 2016; Carreira-Perpinán & Tavallali, 2018) is a binary decision tree whose internal nodes correspond to hyperplanes (not necessarily axis-parallel) and leaf nodes correspond to a label (that is either $+1$ or $-1$). The label for a given instance $\mathbf{x}$ is got by traversing the ODT from root to the leaf. The instance $\mathbf{x}$ goes along the left or right child of an internal node based on which side of its hyperplane $\mathbf{x}$ falls on. A simple 3 level decision tree that we will use as a running example is given in Figure 1.

### 2.1 Failure of Greedy Methods

From Figure 1 one can convince themselves that to classify the given data perfectly, the 7 hyperplanes are essential, and hence a depth 3 ODT is the smallest depth ODT that can express this label function perfectly. If we can search over all 7 hyperplanes simultaneously, using some optimisation

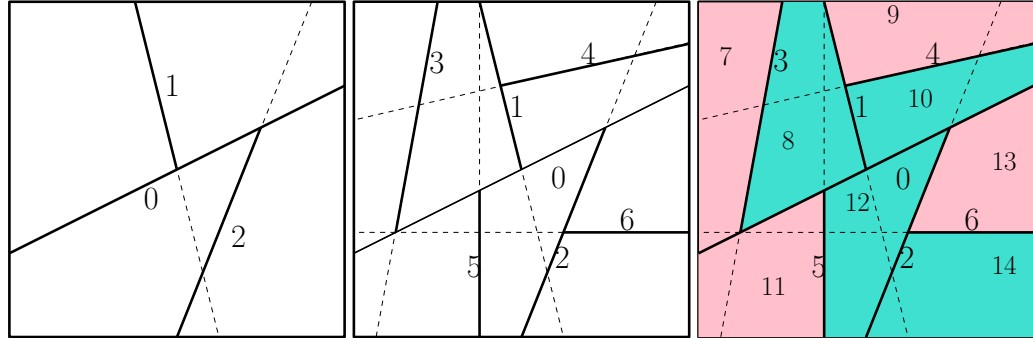

(a) Hyperplanes of root and children in an ODT

(b) Hyperplanes of root, children and grandchildren in an ODT

(c) Leaf node regions are colored based on their label.

Figure 1: Hyperplanes and labelling function for a complete ODT of depth 3. The children of an internal node $i$ are $2i + 1$ and $2i + 2$.

algorithm with enough training data, we would indeed be able to recover the desired ODT. However, most decision tree methods are greedy in nature and cannot do. In particular, for this example, no greedy decision tree algorithm would choose the root node to correspond to the true decision tree root node (indicated by $0$ in the figures). This is because while splitting along the desired line is essential to get the smallest depth decision tree, it is not the best greedy choice. In fact, all known metrics for evaluating splits (accuracy, information gain etc.) would all evaluate the hyperplane corresponding to $0$ in the figure poorly.

This idea can be easily generalised to get a family of such labelling functions : $f^*$ functions corresponding to complete ODTs, whose hyperplanes are oriented in random directions, split the data in each node in a balanced manner and whose leaf node labels are such that sibling leaf nodes do not both get the same label. For any such labelling function $f^*$, all greedy decision tree learning methods fail. Even ODT construction methods that are not purely greedy in nature (Zantedeschi et al., 2020; Bertsimas & Dunn, 2017),(Lee & Jaakkola, 2020) seem to fail for such labeling functions. In fact, as we show in Section 5, even kernel methods which correspond to learning linear models with a fixed feature function fail on such data. Only algorithms that are capable of learning features, such as deep neural networks, perform well on these data.

## 2.2 HYPERPLANE DISCONTINUITY SCORE

We now define a new metric to evaluate ODT splits based on how "discontinuous" the label function is in the vicinity of the hyperplane in question. We define the *hyperplane discontinuity score* (HDS) $\gamma(\mathbf{w}, b)$ for a given distribution $D$ and label function $f^*$, and hyperplane $H = \{\mathbf{x} : \mathbf{w}^\top \mathbf{x} + b = 0\}$ as follows.

$$\gamma(\mathbf{w}, b) = \mathbf{P}_{\mathbf{x} \sim D} \left( f^* \left( \mathbf{x} + \epsilon \frac{\mathbf{w}}{\|\mathbf{w}\|} \right) \neq f^* \left( \mathbf{x} - \epsilon \frac{\mathbf{w}}{\|\mathbf{w}\|} \right) \; \middle| \; \frac{\mathbf{w}^\top \mathbf{x} + b}{\|\mathbf{w}\|} = 0 \right),$$

where $\epsilon > 0$ is a small number. The value of $\gamma(\mathbf{w}, b)$ ranges between 0 and 1. A value of $\gamma(\mathbf{w}, b) = 1$ corresponds to maximum discontinuity, i.e. for all points $\mathbf{x}$ on the hyperplane, pairs of points close to $\mathbf{x}$ lying on opposite sides of the hyperplane have different labels. Similarly, a value of $\gamma(\mathbf{w}, b) = 0$ corresponds to maximum continuity, i.e. for any point $\mathbf{x}$ on the hyperplane, pairs of points close to $\mathbf{x}$ lying on opposite sides of the hyperplane have the same label.

Based on this definition, we can immediately see that for the data in Figure 1, all hyperplanes $\mathbf{w}, b$ except the 7 hyperplanes corresponding to the true decision tree labelling function have a HDS of 0. The seven hyperplanes corresponding to the true decision tree all have a HDS strictly larger than 0. In fact, this is true for any data where the true labelling function $f^*$ corresponds to an oblique decision tree. We also observe that for higher dimensions $d$ when the data is labelled by an ODT, the HDS for hyperplanes corresponding to the internal nodes of the true ODT increases with decrease in distance of the node to the root. This suggests a decision tree construction algorithm which populates its nodes using hyperplanes with large discontinuity scores, where the hyperplane with the largest

HDS goes to the root and so on. A direct implementation of this suffers from computational and robustness issues, however this core idea is quite central to the rest of the paper.

The HDS is quite impractical to compute from finite data due to its dependence on the true label label function $f^*$ and marginal distribution of the instance $D$. However, it can be approximated quite well, by expanding the conditioning criterion to include points $\mathbf{x}$ that are close to the hyperplane, e.g. $\frac{\mathbf{w}^\top \mathbf{x} + b}{\|\mathbf{w}\|} \in [-\frac{\epsilon}{2}, \frac{\epsilon}{2}]$, and not just those exactly on it. With finite training data one is almost sure to not have the true labels for $\mathbf{x} + \epsilon \frac{\mathbf{w}}{\|\mathbf{w}\|}$ and $\mathbf{x} - \epsilon \frac{\mathbf{w}}{\|\mathbf{w}\|}$, however we can relax this by using the label of the nearest neighbours of these points (excluding $\mathbf{x}$). Another practical issue is to ensure that the hyperplane $\{\mathbf{x} : \mathbf{w}^\top \mathbf{x} + b = 0\}$ passes through or divides significant amount of the data. Otherwise, the conditioning in the definition of HDS could become vacuous. We can account for this by only considering hyperplanes $\mathbf{w}, b$ such that $\mathbf{w}^\top \mathbf{x} + b$ is positive (negative) for at least (say) $10\%$ of the data $\mathbf{x}$ drawn from $D$.

While hyperplanes are the main objects of interest in this paper, one can easily extend this idea to general surfaces/manifolds. We conjecture that such surfaces of high discontinuity in the data labelling function are the main drivers of feature learning in deep networks. Note that separating hyperplanes (or surfaces) that separate the data into distinct class labels all have high HDS, but the reverse is not true – one can have hyperplanes of high HDS that do not separate the data (e.g. hyperplane 0 in Figure 1(c).

## 3 DEEP LINEARLY GATED NETWORK(DLGN)

ReLU networks have been a workhorse of deep learning and is the current focus of several theoretical results that aim to explain the success of deep learning over kernel methods. A mainstay of such results is that ReLU networks can learn "features" of the data relevant to the task. While this statement is quite plausible, it is ambiguous due to the vague nature of the term 'features'. There are multiple valid notions of features for deep networks, the main schools of thought on this are given below:

1. The last layer neurons (Daniely, 2017; Lee et al., 2019) are a natural feature choice as the prediction function can simply be viewed as a linear function of the last layer. However, the why, when and how of last layer learning useful/relevant features for the classification task remain impenetrable.

2. The Neural Tangent Kernel (NTK) feature (Jacot et al., 2018; Arora et al., 2019), which corresponds to linearising the neural network prediction function around initialisation. The constant NTK setting simply asserts no feature learning takes place, which is in contrast to empirical results. Tools to study the change of the NTK features during training are an interesting and active area of research. (Atanasov et al., 2022; Baratin et al., 2021; Bordelon et al., 2020; Damian et al., 2022; Chen et al., 2022; Shi et al., 2022; Woodworth et al., 2020; Atanasov et al., 2021; Fort et al., 2020; Hu et al., 2020; Ba et al., 2022; Chizat & Bach, 2018)

3. Directions in activation space (Olah et al., 2020) are an intriguing new possibility and has had practical success in some scenarios like Word2Vec (Church, 2017).

In this paper, we take an alternate route of designing a novel architecture that has the same "feature learning" properties of ReLU networks that enable it to outperform linear and kernel methods. The major difference between this novel architecture and the ReLU network is that it allows for a natural notion of features that is interpretable and decomposable.

### 3.1 ARCHITECTURE DETAILS

The deep linearly gated network also has an architecture similar to that of a ReLU network, and is defined by neurons residing in multiple layers. For simplicity, we assume the architecture consists of $L$ hidden layers with $m$ neurons in each layer. The architecture is parameterized by matrices $W^1, W^2, \ldots, W^L$ and $U^2, \ldots, U^L$, and by vectors $\mathbf{b}^1, \mathbf{b}^2, \ldots, \mathbf{b}^L$ and $\mathbf{u}^1, \mathbf{u}^{L+1}$. The matrices $W^2, \ldots, W^L$ and $U^2, \ldots U^L$ are all of shape $m \times m$. The vectors $\mathbf{b}^1, \ldots \mathbf{b}^L$ are all $m$-dimensional. $W^1$ has shape $m \times d$. $\mathbf{u}^1$ and $\mathbf{u}^{L+1}$ are vectors of size $d$ and $m$ respectively.

Table 1: ReLU network(R) and DLGN(D) test accuracy on CIFAR10 with a simple 5 layer convolutional architecture and also ResNet34 / ResNet110 architectures.

| Conv5(R) | Conv5(D) | Res34(R) | Res34(D) | Res110(R) | Res110(D) |
|----------|----------|----------|----------|-----------|-----------|
| 72.17 | 72.2 | 91 | 86 | 94 | 89 |

The architecture is most naturally described using the notion of paths, which we denote by $\pi = (i_1, \ldots, i_L) \in [m]^L$, giving the sequence of hidden nodes that the path consists of. Let $\Pi = [m]^L$ denote the set of all paths. The output of the model is given as follows:

$$\widehat{y}(\mathbf{x}) = \sum_{\pi \in \Pi} g_\pi f_\pi(\mathbf{x}) \tag{1}$$

where $f_\pi$ is called the path gating function corresponding to path $\pi$ and $g_\pi$ is the value of path $\pi$. The path gating function $f_\pi$ is defined by (what we call) the gating network – a deep linear network with weights $W^1, \ldots, W^L$ and biases $\mathbf{b}^1, \ldots \mathbf{b}^L$. The path gating function $f_\pi$ can be decomposed as the product of individual neuron gating functions that make up the path $\pi$. The path gating function for path $\pi = (i_1, \ldots, i_L)$ is defined as follows,

$$f_\pi(\mathbf{x}) = \prod_{\ell=1}^{L} \mathbf{1}\left(\boldsymbol{\eta}_{i_\ell}^\ell(\mathbf{x}) \geq 0\right) \tag{2}$$

$$\forall \ell \in [L], \qquad \boldsymbol{\eta}^\ell(\mathbf{x}) = W^\ell \boldsymbol{\eta}^{\ell-1}(\mathbf{x}) + \mathbf{b}^\ell = V^\ell \mathbf{x} + \mathbf{c}^\ell \tag{3}$$

where $\boldsymbol{\eta}^0(\mathbf{x}) = \mathbf{x}$ and $\forall \ell \in [L]$ the matrices $V^\ell \in \mathbb{R}^{m \times d}$ and vectors $\mathbf{c}^\ell \in \mathbb{R}^m$ form the 'effective' weights and biases of the neurons in layer $\ell$ and are given as $V^\ell = W^\ell W^{\ell-1} \ldots W^1$ and $\mathbf{c}^\ell = b^\ell + W^\ell \mathbf{c}^{\ell-1}$ with $\mathbf{c}^0 = \mathbf{0} \in \mathbb{R}^d$.

The value $g_\pi$ of a path $\pi = (i_1, \ldots, i_L)$ is also defined by a network (that we call) the value network – a deep linear network with weights $U^2, \ldots, U^L, \mathbf{u}^{L+1}$, no biases, and input given by $\mathbf{u}^1$. It is simply the product of weights along the path $\pi$.

$$g_\pi = \mathbf{u}_{i_1}^1 \left[\prod_{\ell=2}^{L} U_{i_\ell, i_{\ell-1}}^\ell\right] \mathbf{u}_{i_L}^{L+1} \tag{4}$$

The model as defined in Equation (1) seems computationally hard to implement in a forward pass, but due to standard matrix multiplication properties can be easily implemented at a cost that is less than twice the cost of a ReLU net with the same $mL$ hidden nodes. i.e.

$$\widehat{y}(\mathbf{x}) = \langle \mathbf{u}^{L+1}, h^L(\mathbf{x}) \rangle$$

where $h^1(\mathbf{x}) = \mathbf{1}(\boldsymbol{\eta}^1(\mathbf{x}) \geq 0) \circ \mathbf{u}^1$ and $h^\ell(\mathbf{x}) = \mathbf{1}(\boldsymbol{\eta}^\ell(\mathbf{x}) \geq 0) \circ \left(U^\ell h^{\ell-1}(\mathbf{x})\right)$ for $\ell > 1$. The gates $\boldsymbol{\eta}$ are as defined in Equation 3. The symbol $\circ$ represents elementwise multiplication. In order to learn the gating function parameters and back-propagate the gradient to $W$ and $\mathbf{b}$, we will need to replace the indicator function by a sigmoid. i.e. we replace $\mathbf{1}(a \geq 0)$ with $\sigma(\beta a)$ where $\sigma$ is the standard sigmoid function and $\beta > 0$ is a hyperparameter.

Thus the model $\widehat{y}$ is defined as a linear combination of path gating functions. The natural rigorous choice for features in the DLGN model would be the path gating functions $f_\pi$, but they are exponentially many in number. Fortunately, however the path gating functions decompose further into product of neuron gating functions which are indicator functions over half spaces. These are much simpler, number only a total of $mL$ and make an apt choice of elemental features.

The DLGN model is more easily amenable to be broken down into easily understood components than the ReLU network. It also outperforms fixed feature methods like kernel methods and is only marginally inferior to ReLU networks in real data tasks. See Table 1 for a test accuracy comparison between DLGN and ReLU networks on the CIFAR10 dataset.

### 3.2 PROPERTIES OF A TRAINED DLGN

A trained DLGN shows some interesting properties, that is not possible to even check on ReLU networks. The most important of these properties is a tendency of the effective hyperplane of the gating

Table 2: Number of DLGN hyperplanes (after training) within a given distance of the label function ODT hyperplanes. At initialisation all these numbers are equal to zero. The 15 ODT internal nodes are numbered 0 to 14, with 0 as root.

| Distance | 0 | 1 | 2 | 3 | 4 | 5 | 6 | 7 | 8 | 9 | 10 | 11 | 12 | 13 | 14 |
|---|---|---|---|---|---|---|---|---|---|---|---|---|---|---|---|
| 0.1 | 31 | 4 | 4 | 0 | 0 | 0 | 0 | 0 | 0 | 0 | 0 | 1 | 2 | 1 | 3 |
| 0.2 | 162 | 8 | 18 | 0 | 0 | 2 | 0 | 2 | 1 | 5 | 2 | 2 | 6 | 2 | 6 |
| 0.3 | 284 | 14 | 40 | 1 | 2 | 5 | 2 | 2 | 1 | 6 | 5 | 2 | 8 | 4 | 9 |

neurons given by $V_i^\ell$ and $\mathbf{c}_i^\ell$ to cluster around hyperplanes of discontinuity in the label function. This is most easily seen in data where the true label function corresponds to an ODT.

Figure 2 illustrates an example scenario when a 3-hidden layer DLGN is trained on data given in Figure 1(c). The initial hyperplanes given by $V^1, \mathbf{c}^1$ and $V^2, \mathbf{c}^2$ and $V^3, \mathbf{c}^3$ as shown in Figure 2(a-c) are essentially random. However, after training the hyperplanes in the later layers show a remarkable tendency to move towards the hyperplanes corresponding to the decision tree – particularly that of nodes close to the root (See Figures 2(d-f)).

Do note that Figure 2 is a schematic illustration and the decision tree hyperplane seeking behaviour of the DLGN hyperplanes are exaggerated to illustrate the idea properly. Table 2 gives the results of a real experiment on synthetic dataset in which the data is a 19-dimensional vector labelled by a depth-4 complete ODT with 15 internal nodes. A 4 hidden layer DLGN with 1000 neurons in each layer was trained on this dataset containing about 18000 data points. For each node in the ODT, we count the number of DLGN hyperplanes within a distance of $0.1, 0.2$ and $0.3$ from it. We define the distance between two hyperplanes $H(\mathbf{v}, c) = \{\mathbf{x} : \mathbf{v}^\top \mathbf{x} + c = 0\}$ and $H(\mathbf{z}, d) = \{\mathbf{x} : \mathbf{z}^\top \mathbf{x} + d = 0\}$ as

$$\text{dist}(\mathbf{v}, \mathbf{z}) = \min \left( \left\| \frac{\mathbf{z}}{\|\mathbf{z}\|} - \frac{\mathbf{v}}{\|\mathbf{v}\|} \right\|, \left\| \frac{\mathbf{z}}{\|\mathbf{z}\|} + \frac{\mathbf{v}}{\|\mathbf{v}\|} \right\| \right)$$

We ignore the scalars $c$ and $d$ for convenience, but they can also be incorporated in a more sophisticated distance function for hyperplanes. We normalise the vectors $\mathbf{z}$ and $\mathbf{v}$ before finding the distance because scalar multiples of the coefficients do not change the hyperplanes. We also identify $\mathbf{v}$ with $-\mathbf{v}$ for the same reason. For large $d$, most pairs of vectors are orthogonal to each other, and hence a typical value of $\text{dist}(\mathbf{v}, \mathbf{z})$ is about $\sqrt{2}$. Thus, at initialisation there are almost no DLGN hyperplanes close (say distance less than $0.3$) to any of the ODT hyperplanes. But it can be clearly seen from Table 2, that the training process attracts the DLGN hyperplanes towards the ODT hyperplanes. This happens most notably for the root node, despite the root node having almost zero information gain or accuracy increase in the greedy decision tree construction setting.

We conjecture that there is a fundamental principle of feature learning in deep networks at play here, and studying the reason for this behaviour is a very interesting direction that is beyond the scope of this paper. For the purpose of this paper, we loosely assume the following principle: " *The model features seek to match with the manifolds of discontinuity in the label function during neural network training*". For the case of DLGN models and ODT labelling functions, this reduces to saying that the DLGN hyperplanes move towards the ODT hyperplanes, with a preference towards the close-to-root node hyperplanes which have a higher HDS.

## 4 DLGN BASED DECISION TREE CONSTRUCTION

Based on the principle conjectured in the previous section, we devise a novel decision tree learning algorithm as follows. Figure 3 gives a schematic illustration of the algorithm. In the first stage, a DLGN is trained on the entire data. Our conjectured principle would then imply that the hyperplane with the highest HDS would attract much more DLGN hyperplanes than others resulting in a detectable cluster around it. A clustering is performed over the learned DLGN hyperplanes and the largest cluster is chosen (See Figure 3(a)). The cluster center corresponding to it is chosen as the root node in the decision tree construction procedure. Based on the root node hyperplane, the

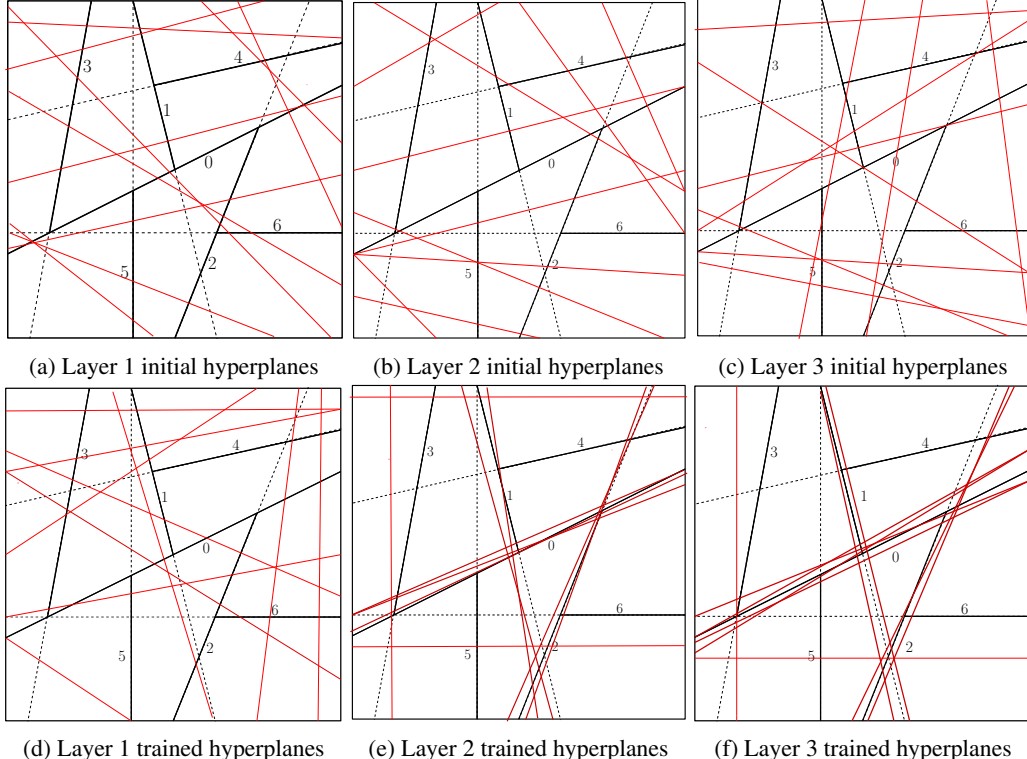

| (a) Layer 1 initial hyperplanes | (b) Layer 2 initial hyperplanes | (c) Layer 3 initial hyperplanes |
| --- | --- | --- |
| (d) Layer 1 trained hyperplanes | (e) Layer 2 trained hyperplanes | (f) Layer 3 trained hyperplanes |

Figure 2: An illustration of DLGN hyperplanes before and after training on data in Figure 1c.

training data can be split into two halves and the procedure can be repeated on both halves recursively (See Figure 3(b,c)) until the data for training becomes too small or contains only one class. A oblique decision tree can thus be constructed by incorporating the largest cluster centers of the trained DLGN hyperplanes in the appropriate nodes of a tree.

The details of the above procedure is given in Algorithm 1. It returns a decision tree consisting of internal nodes and leaf nodes. Internal nodes are represented by a hyperplane and pointers to two child nodes. Leaf nodes are represented by a value that is either $+1$ or $-1$.

The key subroutine in Algorithm 1 is the FINDDISCONTHYPERPLANE function detailed in Algorithm 2. It trains a DLGN on a classification dataset, clusters the DLGN hyperplanes and splits the data based on this hyperplane. In our experiments we used the DBScan (Ester et al., 1996) algorithm for clustering the DLGN hyperplanes as it is robust to outliers.

---

**Algorithm 1** Building a decision tree from trained DLGN

---

**Arguments:** Binary classification training set containing pairs $(\mathbf{x}_i, y_i)$
**Outputs:** Root of an ODT

1: **function** BUILDTREE(currdata)
2:    **if** IMPURE(currdata) **or** LARGE(currdata) **then**
3:       dataleft,dataright,$\mathbf{v}^*$,$\mathbf{c}^*$ ← FINDDISCONTHYPERPLANE(currdata)
4:       leftST ← BUILDTREE(dataleft)
5:       rightST ← BUILDTREE(dataright)
6:       **return** NODE($\mathbf{v}^*, \mathbf{c}^*$, leftST, rightST)
7:    **end if**
8:    **lv** ← MAJORITYLABEL(currdata)
9:    **return** NODE(*value*=**lv**)
10: **end function**

---

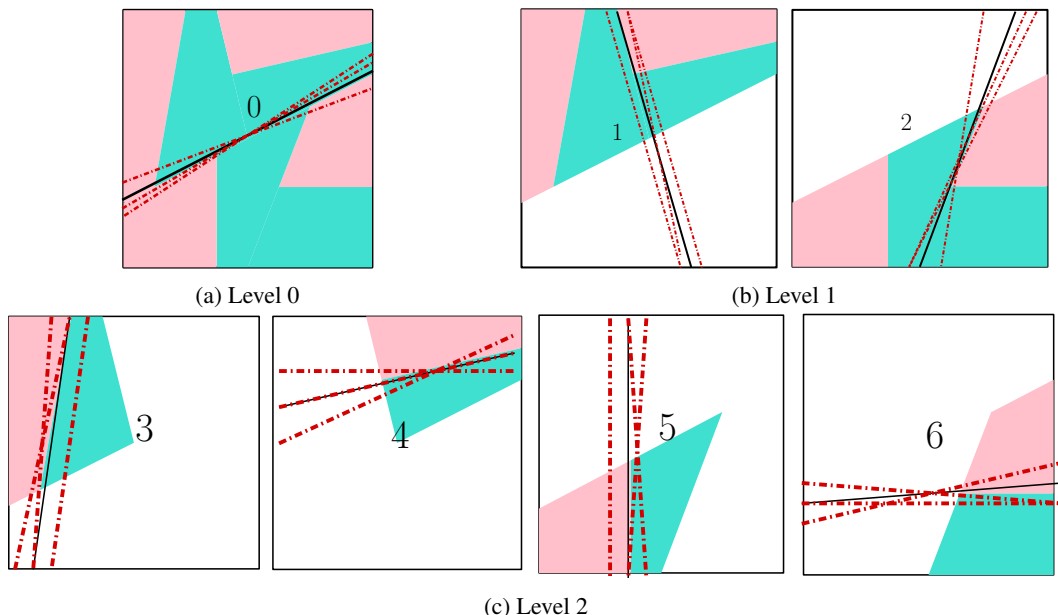

(a) Level 0 (b) Level 1

(c) Level 2

Figure 3: DLGN based ODT Learning. Illustration of the recursive procedure where for each level starting at 0, we have $2^{\text{level}}$ DLGNs are trained on different splits of the data given by previous levels. The main result of each training run is the largest cluster of the learned DLGN hyperplanes, which is shown in the figures by red dashed lines.

---

**Algorithm 2** Finding Discontinuous Hyperplane

---

**Arguments:** Binary classification training set containing pairs $(\mathbf{x}_i, y_i)$
**Outputs:** Data split into 2 halves along a hyperplane, and the hyperplane parameters $\mathbf{w}, \mathbf{b}$
 1: **function** FINDDISCONTHYPERPLANE(`currdata`)
 2:     `model` $\leftarrow$ TRAINDLGN(`currdata`)
 3:     $V, \mathbf{c} \leftarrow$ RETURNGATEHYPERPLANES(`model`)
 4:     $\mathbf{v}^*, \mathbf{c}^* \leftarrow$ LARGESTCLUSTERCENTER($V, \mathbf{c}$)
 5:     `dataleft, dataright` $\leftarrow$ SPLITDATA(`currdata`,$\mathbf{v}^*, \mathbf{c}^*$)
 6:     **return** `dataleft, dataright`,$\mathbf{v}^*, \mathbf{c}^*$
 7: **end function**

---

**Algorithm 3** Return gates of a trained DLGN model

---

**Arguments:** A DLGN model (Parameters: $W^1, \ldots, W^L, \mathbf{b}^1, \ldots, \mathbf{b}^L, \mathbf{u}^1, U^2, \ldots, U^L, \mathbf{u}^{L+1}$)
**Outputs:** $mL$ hyperplanes in the input dimension
 1: **function** RETURNGATEHYPERPLANES(`model`)
 2:     $\mathbf{c}^0 = \mathbf{0}$
 3:     **for** $l \leftarrow 1$ **to** $L$ **do**
 4:         $V^\ell \leftarrow W^\ell W^{\ell-1} \ldots W^1$
 5:         $\mathbf{c}^\ell \leftarrow b^\ell + W^\ell \mathbf{c}^{\ell-1}$
 6:     **end for**
 7:     $V \leftarrow V^1, \ldots, V^L$
 8:     $\mathbf{c} \leftarrow \mathbf{c}^1, \ldots, \mathbf{c}^L$
 9:     **return** $V, \mathbf{c}$
10: **end function**

---

Table 3: Dataset and model statistics

| Dataset | Train size | Features count | DLGN Arch. | DLGN DT Arch. |
|---|---|---|---|---|
| SD - I | 18k | 19 | 1000×4 | 500×3,d=3 |
| SD - II | 24.5k | 100 | 500×3 | 500×3,d=2 |
| SD - III | 45.4k | 500 | 500×3 | 500×3,d=2 |
| Adult | 29.3k | 14 | 125×4 | 750×3,d=6 |
| Bank | 27.12k | 16 | 1000×3 | 750×3,d=6 |
| Credit card | 18k | 23 | 200×3 | 750×3,d=6 |
| Telescope | 11.4k | 10 | 10×3 | 750×3,d=6 |
| Rice | 2.3k | 7 | 2×4 | 750×3,d=6 |
| Statlog | .1k | 20 | 100×5 | 750×3,d=1 |
| Spambase | 2.8k | 57 | 200×4 | 750×4,d=1 |
| Gyro | 19.2k | 8 | 125×4 | 750×3,d=6 |
| Swarm | 14.4k | 2400 | 200×3 | 750×3,d=6 |

Table 4: Test accuracy on synthetic datasets

| Dataset | DLGN | ReLU | SVM | CART | RF | SDT | Zan DT | DLGN DT |
|---|---|---|---|---|---|---|---|---|
| SD - I | 98.6 | 98.9 | 83.5 | 61.3 | 72.6 | 93.4 | 89.4 | 97.1 |
| SD - II | 99.3 | 94.8 | 73.9 | 54.6 | 65.5 | 96.7 | 89.7 | 97.8 |
| SD - III | 93.7 | 72.0 | 68.0 | 51.3 | 60.6 | 88.4 | 67.0 | 94.2 |

## 5 EXPERIMENTS

We assessed the performance of our methods, DLGN and DLGN-aided decision trees(DLGN_DT), in comparison to standard algorithms (multilayer ReLU networks and Kernel SVMs) . The evaluation was conducted on Synthetic Datasets(SD) and real datasets, with the details of the dataset and model architectures outlined in Table 3. DLGN_Arch. denotes the architecture yielding the reported accuracy for the DLGN model chosen via validation. We used Zan DT Zantedeschi et al. (2020) as a standard oblique tree baseline method. Other baseline algorithms include the standard random forest (which has standard axis parallel decision trees) and the Soft Decision Tree Frosst & Hinton (2017).

### 5.1 DISCUSSION OF PERFORMANCE ON SYNTHETIC DATASETS

We construct 3 different synthetic datasets, where the data $\mathbf{x}$ was drawn uniformly from $[-1,1]^d$ and the labelling function is a complete ODT with internal node hyperplane normals drawn independently from a circularly symmetric distribution, and biases chosen such that the data is split almost equally among the children. The leaf node labels are chosen such that sibling labels get opposite signs. The three datasets (SD-I, SD-II and SD-III) have different input dimensions (19, 100 and 500 respectively) and use decision trees with different depths (4,3 and 3 respectively). The results on synthetic datasets are summarized in Table 4. DLGN and DLGN_DT outperformed both standard ML methods like SVM, CART and other oblique decision tree methods like Zan DT. Soft Decision Trees perform comparably to DLGN-DT for lower dimensional data (SD-I and SD-II) but underperform DLGN-DT on higher dimensional data. Standard ReLU nets give good accuracy for several synthetic datasets, but seem to require extensive hyperparameter tuning , particularly for higher dimensional data. Another qualitative property of DLGN-DT that is not fully reflected in Table 4 is that, the hyperplanes in the learned tree very closely match the hyperplanes in the true ODT.

### 5.2 DISCUSSION OF PERFORMANCE ON UCI DATASETS

We also conducted experiments on real UCI datasets. DLGN exhibited superior or comparable performance over other ML algorithms. DLGN_DT demonstrated comparable accuracy to other decision tree learning algorithms. Detailed results can be found in Table 5.

Table 5: Test accuracy on UCI datasets

| Dataset | DLGN | ReLU | SVM | CART | RF | SDT | Zan DT | DLGN DT |
|---|---|---|---|---|---|---|---|---|
| Adult | 85.3 | 83.2 | 84.4 | 83.2 | 86.4 | 84.8 | 84.6 | 84.1 |
| Bank | 91.7 | 89.0 | 90.6 | 90.0 | 91.3 | 91.5 | 91.2 | 90.6 |
| Credit card | 81.6 | 78.0 | 81.2 | 77.8 | 81.2 | 81.4 | 81.1 | 81.1 |
| Telescope | 88.1 | 87.7 | 87.0 | 82.9 | 88.1 | 86.4 | 87.1 | 85.6 |
| Rice | 92.8 | 92.4 | 91.7 | 91.9 | 91.9 | 92.5 | 92.2 | 92.7 |
| Statlog | 77.5 | 72 | 71.5 | 65.5 | 77.5 | 75.0 | 75.5 | 80.0 |
| Spambase | 94.1 | 94 | 93.1 | 89.1 | 95.0 | 93.4 | 93.3 | 93.6 |
| Gyro | 98.8 | 98.6 | 98.3 | 98.7 | 99.3 | 98.0 | 98.6 | 98.1 |
| Swarm | 100 | 100 | 100 | 99.8 | 100 | 100 | 100 | 99.9 |

## 6 CONCLUSION

We devised a novel architecture, with interpretable 'features' that show an interesting property of seeking out discontinuities in the label function. We exploited this property to construct a novel decision tree learning algorithm that performs well even in scenarios where greedy methods fail. Studying the DLGN architecture further to characterise this feature learning phenomenon is an interesting direction of future work.

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

# A APPENDIX

## A.1 EXPERIMENTAL SETUP

The results presented in Table 1 were obtained using the CIFAR-10 dataset. Conv5(R) denotes a five-layered Convolutional Neural Network (CNN) employing Rectified Linear Unit (ReLU) activation, while Conv(D) signifies a five-layered Convolutional model featuring DLGN architecture instead of ReLU. The subsequent section outlines the specific hyperparameters and configurations utilized for this analysis.

### A.1.1 DEATAILS OF EXPERIMENTS PERFORMED ON CIFAR10

### A.1.2 HYPERPARAMETERS

- Number of Convolution Layers = 5
- Number of filters in each layer = 26
- Optimizer: Adam
- Learning rate = 2e-4

### A.1.3 ARCHITECTURES

- ReLU network : 5 Convolution layers with ReLU activation in each layer, followed by Global Average Pooling, followed by 1 Dense layer with 64 neurons.
- DLGN: 5 Convolution layers, followed by Global Average Pooling, followed by 1 Dense layer with 64 neurons.

## A.2 DETAILS OF EXPERIMENTS PERFORMED ON SYNTHETIC DATASETS

The results are presented in Table 4. This dataset is synthetically generated from an oblique binary tree with specified dimensions, depth, and a defined number of data points. The decision tree structure is constructed, and random weights for each internal node are generated. Data points are then assigned labels based on their position in the tree. The resulting dataset is pruned to remove points that pass through hyperplanes by **threshold** parameter. The final output includes the pruned data, labels, and information about the tree's structure. Three synthetic datasets (SD) are used, named SD - I, SD - II, and SD - III.
The parameters used for constructing the datasets are as follows:

**Synthetic Dataset I (SD - I):**

- Number of datapoints = 300000
- Dimension of the data = 19
- Seed: 1387
- Depth = 4
- threshold = 0.1

**Synthetic Dataset II (SD - II):**

- Number of datapoints = 100000
- Dimension of the data = 100
- Seed: 1387
- Depth = 3
- threshold = 0.05

**Synthetic Dataset III (SD - III):**

- Number of datapoints = 100000

- Dimension of the data = 500

- Seed: 365

- Depth = 3

- threshold = 0.01

The data and model(DLGN and DLGN_DT) statistics are given in Table 3

### A.2.1  HYPERPARAMETERS OF DLGN MODEL

- Number of hidden layers = 3(SD-II,SD-III) 4(SD-I)

- Number of nodes in each layer = 1000(SD-I) 500(SD-II,SD-III)

- Optimizer: Adam

- Learning rate = 0.001

- beta=3

### A.2.2  HYPERPARAMETERS OF DLGN DT MODEL

**DLGN Model used:**

- Number of hidden layers = 3

- Number of nodes in each layer = 500

- Optimizer: Adam

- Learning rate = 0.001

- beta=3

**Clustering Model used:** We are using DBScan clustering to find the largest cluster center and towards the bottom of the tree in case no cluster is detected, we are using Logistic Regression to find the features for the nodes.

- Name of the algorithm = DBScan

- eps = 0.2

- **min_samples:**
  **SD-I,SD-II**

    - Node 0 - 10
    - Node 1 - 15
    - Node 2 - 30
    - Other nodes - 40

  **SD-III**

    - Node 0 - 30
    - Node 1 - 100
    - Node 2 - 50
    - Other nodes - 100

### A.3  DETAILS OF EXPERIMENTS PERFORMED ON UCI DATASETS

The results are presented in Table 5.The UCI Machine Learning Repository is a collection of numerous datasets for machine learning tasks. We picked our real datasets from the UCI dataset repository

available at `https://archive.ics.uci.edu/datasets`. The datasets picked for our binary classification tasks are **Adult**[2] **Bank**[3] **Credit card**[4] **Telescope**[5] **Rice**[6] **Statlog**[7] **Spambase**[8] **Gyro**[9] **Swarm**[10]

The data and model statistics are given in Table 3:

### A.3.1 HYPERPARAMETERS OF DLGN MODEL

- Optimizer: Adam
- Learning rate = 0.001
- beta=3

### A.3.2 HYPERPARAMETERS OF DLGN DT MODEL

**Clustering Model used:** We are using DBScan clustering to find the largest cluster center and towards the bottom of the tree in case no cluster is detected, we are using Logistic Regression to find the features for the nodes.

- Name of the algorithm = DBScan
- eps = 0.2 (Spambase) 0.3(rest all)
- **min_samples:**
    - Adult - 5
    - Bank - 5
    - Credit Card- 6
    - Telescope - 6
    - Rice - 6
    - Statlog - 5
    - Spambase Card- 10
    - Gyro - 6
    - Swarm - 6

### A.4 ALGORITHMS USED

---

[2]`https://archive.ics.uci.edu/dataset/2/adult`

[3]`https://archive.ics.uci.edu/dataset/222/bank+marketing`

[4]`https://archive.ics.uci.edu/dataset/350/default+of+credit+card+clients`

[5]`https://archive.ics.uci.edu/dataset/159/magic+gamma+telescope`

[6]`https://archive.ics.uci.edu/dataset/545/rice+cammeo+and+osmancik`

[7]`http://archive.ics.uci.edu/dataset/144/statlog+german+credit+data`

[8]`http://archive.ics.uci.edu/dataset/94/spambase`

[9]`https://archive-beta.ics.uci.edu/dataset/755/accelerometer+gyro+mobile+phone+dataset`

[10]`https://archive.ics.uci.edu/dataset/524/swarm+behaviour`

