# OpenReview forum: "Enhancing Decision Tree Learning with Deep Networks"
_ICLR.cc/2024/Conference — Submitted to ICLR 2024_

### Official Review · Reviewer_ugFu · 2023-10-19

**Soundness:** 2 fair
**Presentation:** 2 fair
**Contribution:** 2 fair
**Rating:** 3
**Confidence:** 4

**Summary:**

The paper presents a decision tree construction algorithm that outperforms traditional methods, especially when dealing with uncorrelated root nodes. It also offers insights into the inner workings of deep neural networks' feature learning mechanisms.

**Strengths:**

The paper is easy to follow. I believe that providing an intuitive visualization of decision boundaries and explanations using figures, such as the algorithm diagram in Figure 3, would be helpful support for readers.

**Weaknesses:**

I believe that constructing a greedy decision tree offers significant advantages in terms of computational time. While it is possible to make the search more complex, I think there is a deliberate choice not to create overly complicated trees in order to balance computation time and performance. In this sense, it seems that the proposed method involves complex processing during tree construction, but there is no evaluation of the computational cost incurred in doing so. I think it's necessary to have a diverse range of evaluations from perspectives other than just accuracy in order to assess the usefulness of the proposed approach.

Furthermore, since the connection between oblique trees and ReLU networks has been extensively studied, it is necessary to clarify their comparison, mention in related work, and the differences in their respective positions.

When presenting experimental results such as in Table 1, please evaluate the errors.

The mention "Even ODT construction methods that are not purely greedy in nature seem to fail for such labeling functions" is present in the text, but it appears that there is no supporting experimental or background information for this assertion.

**Questions:**

1: Please provide information about the training time (Check the weaknesses part).

2: I imagine that when using a single decision tree, one may not prioritize accuracy too much. If you want to push for higher accuracy, it's natural to adopt approaches that use multiple trees like Random Forest or Gradient Boosting Decision Trees. However, other factors such as interpretability and processing speed for a single tree might be important. Are there any benefits from that perspective?

3: Section 3.2 contains the mention: "A trained DLGN shows some interesting properties that are not possible to even check on ReLU networks." However, it is well-known that ReLU networks partition feature space linearly. In that sense, I believe hyperplanes can be checked, can't they? (e.g., “Neural Networks are Decision Trees, Caglar Aytekin, (2022)”)

---

> ### Author Response · Authors · 2023-11-21
> **ReLU networks**
>
> Weaknesses:
>
> The connection between oblique trees and ODTs are well studied, but mostly in an "existence" kind of way -- i.e. that there exists an ODT for every ReLU net and vice versa. Not many constructive/analysis results can be found, to the best of our knowledge. For example, we cannot even look at an ODT and a ReLU network and say if they compute the same function, without exhaustively evaluating across the domain.
>
> With a DLGN it is at least possible to check if the hyperplanes of discontinuity in the model (the mL hyperplanes corresponding to the neurons) include the hyperplanes in the ODT. e.g. Table 2 in the paper.
>
> The statement "ODT construction methods that are not purely greedy in nature seem to fail for such labeling functions" is mainly supported empirically by our results on ZanDT (and the newer SDT results). As of yet, we do not know the reasons why these methods  also underperform on the synthetic datasets as we only have a procedural understanding of these algorithms. But we do observe that in addition to lower accuracy, these also fail to retrieve the exact hyperplanes corresponding to the true ODT labelling function -- as opposed to DLGN-DT which retrieves the hyperplanes.
>
> Questions:
>
> 1. Training time questions are answered in the common comments above.
>
> 2. Random Forests etc. also do not perform particularly well on the synthetic data we construct. This shows that "feature learning" is essential. Without capturing the hyperplanes of the true ODT it is not possible to generalize well.
>
> 3. Yes, ReLU networks partition the feature space into polyhedra. But a full breakdown requires 2^{total number of hidden neurons} polyhedrons. While it has been conjectured (Rolnick21) that a vast majority of them are empty, apriori it is impossible to know which are empty and which are not. And it is also not possible to break down these exponentially many regions as simple interactions arising from a much smaller elementary set. DLGNs on the other hand are immensely more graspable -- each neuron is just a half-space. The power of the feature space comes from the intersection of these half-spaces. A much larger set (still only m^L as aopposed to 2^{mL} for ReLU nets) that can be viewed as composed of L (out of mL) elemental half-spaces.

---

### Official Review · Reviewer_67Vr · 2023-10-29

**Soundness:** 2 fair
**Presentation:** 3 good
**Contribution:** 2 fair
**Rating:** 5
**Confidence:** 4

**Summary:**

The paper identifies a family of labelling functions that can be efficiently represented by an oblique decision trees, however existing learning algorithms fail to learn these trees. To overcome this, the paper presents a new splitting criterion (HDS) and present a deep architecture called DLGN that can be used to detect hyperplanes with low HDS to be selected as splits for the internal nodes of the oblique tree.

**Strengths:**

Strengths:
- Interesting and seemingly novel intuition/observation that is represented by the proposed hyperplane discontinuity score
- Experiments seem to support hypothesis on synthetically constructed datasets
- Generally well-written with useful illustrative figures

**Weaknesses:**

Weaknesses:
- The main intuition behind the proposed approach is not established theoretically. Further, even the hypothesis itself is not mathematically and precisely formalized. It seems to be motivated by a specific synthetic construction that is not clear if this construction tends to appears in real problems.
- The empirical support for the main claim (e.g., Table 2) is also based on experiments with synthetic data
- Experimental results for the proposed decision tree construction method are not very convincing: The baseline Zan DT does better on real datasets and outperforms DLGN DT in 5 datasets while DLGN DT outperforms Zan DT in only 3 datasets.
- The experiments could benefit from experiments with additional baselines for oblique decision trees (e.g., TAO [Carreira-Perpinan & Tavallali, 2018] and others mentioned), as well as reporting results on training accuracy.
- Also, there is no discussion or results on the differences in terms of computational resources (the proposed approach seems to require training a neural network in each node of the tree and running DBSCAN on the whole dataset which may hinder the scalability of the approach)
- No discussion if/how this can be extended beyond binary classification


Minor typos, inconsistencies:
- space before "Krishnan et al." page 2
- notation: it looks like $\gamma$ should be parameterized by D and f* as well

**Questions:**

I would appreciate the authors' response to the main weaknesses listed above

---

> ### Author Response · Authors · 2023-11-21
> **Experimental results**
>
> As correctly pointed out, the main claim is captured in Table 2 of the paper. It is one of the first known results where Feature learning is demonstrated directly, and not just inferred via a statement like "Test accuracy improved because of feature learning". (The paper by Chizat and Bach on optimal transport with 1-Layer ReLU nets is another prominent example).
>
> While the DLGN is indeed a much more tractable architecture than the ReLU network, we anticipate that capturing the dynamics of this architecture will still be quite complicated. Case in point: Deep linear networks, which are simply a reparametrised linear model, are still an active area of study for analysing properties of learning.
>
> Theorems, if any, in this setting are going to be quite heavy on assumptions. But a result like Table 2, gives a direction that future papers can try to prove at least in a restricted setting.
>
> The computational concerns and baseline comparisons are addressed in the common comment.
>
> While we show results only on 3 synthetic datasets (where data is labelled by an unknown ODT), we observe similar results in all the synthetic datasets we tried. But they add nothing more than the three datasets studied here, corresponding to small, medium and high-dimensional input data.

---

> > ### Comment · Reviewer_67Vr · 2023-11-23
> >
> > Thank you for your response. I think the lack of theoretical results and the fact that the advantage of the proposed method appears primarily in synthetic datasets remain the main weaknesses.

---

### Official Review · Reviewer_cuTu · 2023-10-30

**Soundness:** 2 fair
**Presentation:** 2 fair
**Contribution:** 1 poor
**Rating:** 3
**Confidence:** 5

**Summary:**

The provided paper introduces an oblique tree learning algorithm that integrates neural networks into its framework. This methodology adheres to a top-down approach in tree construction, where, at each split, a neural network training is employed to separate two classes (thus, applicable to binary classification only). Subsequently, a clustering algorithm is executed to extract a hyperplane from the trained neural network. This hyperplane then serves as the basis for partitioning the data into two subsets, initiating a recursive progression of the algorithm from that point onward.

To evaluate the efficacy and performance of this algorithm, experiments are conducted across various benchmarks, employing several baselines.

**Strengths:**

- the method is easy to understand and implement;
- the same for the paper, easy to follow.

**Weaknesses:**

1. In Section 2.1, when asserting that "all greedy methods would fail," it is essential to state the underlying assumptions supporting this claim. As it stands, I find it challenging to ascertain the veracity of this statement. Consider the dataset below consisting of 2 points (for simplicity):

  x | o

where x and o are data points and "|" represents the decision boundaries. Any greedy split will find | as a solution...

If this proposition is intended to be presented as a theorem, then it necessitates a rigorous formulation and a subsequent proof to establish its validity. It is crucial to uphold the highest standards of mathematical rigor when making such assertions, ensuring that they are substantiated by sound theoretical foundations.

2. **Novelty**. The method resembles soft decision trees (SDTs) [1-3] in its formulation in section 3.1. However, instead of learning hyperplane at each node, the method first fits a NN followed by clustering-based heuristics. This is a bit different since it relies on greedy tree growing procedure. However, similar "neural" tree growing technique (without clustering) was employed in Guo and Gelfand (1992). Here, the method applies "postprocessing" to transform deep NN into hyperplane.

3. **Experiments**. The experiment, as presently conducted, exhibits a notable gap in its evaluation methodology. It notably lacks a comparative analysis against well-established oblique tree learning methods, including those referenced in citations [1-5], as well as the work by Carreira-Perpinan and Tavallali from 2018. Such a comparative assessment is paramount in validating the efficacy and distinctiveness of the proposed approach.

4. The method as is only applicable to binary classification and extending it seems to be nontrivial (except, maybe, one-vs-all)?

---------------

[1] Jordan, M. I. and Jacobs, R. A. (1994). Hierarchical mixtures of experts and the EM algorithm. Neural Computation, 6(2):181–214

[2] Frosst, N. and Hinton, G. (2017). Distilling a neural network into a soft decision tree. arXiv:1711.09784

[3] Hazimeh, H., Ponomareva, N., Mol, P., Tan, Z., and Mazumder, R. (2020). The tree ensemble layer: Differentiability meets conditional computation. In Daumé III, H. and Singh, A., editors, Proc. of the 37th Int. Conf. Machine Learning (ICML 2020).

[4] Zharmagambetov, A., Hada, S. S., Gabidolla, M., and Carreira-Perpiñán, M. Á. (2021b). Non-greedy algorithms for decision tree optimization: An experimental comparison. In Int. J. Conf. Neural Networks(IJCNN’21).

[5] One possible SDT implementation: https://github.com/xuyxu/Soft-Decision-Tree

**Questions:**

- What is Zan DT method? I don't see any references to it...

---

> ### Author Response · Authors · 2023-11-21
> **Failure of Greedy Methods**
>
> Thanks for the comments.
>
> Some of your other concerns were addressed in the common comment.
>
> When we say "all greedy methods would fail" we mean the setting where the labels are coming from an ODT with high (>5) dimensional input and balanced splits in each nodes, with the node hyperplanes being close to orthogonal to other nodes.
>
> e.g. a depth 3 ODT over (say) 10-d data, with each node i going qi.x >0 ? where q1, q2, ..., q_7 are all orthogonal vectors in 10-d space. The 8 leaf nodes are labelled such that siblings have opposite labels. Here no greedy procedure for picking the root node would pick q1.x as the split function. Because the information gain would be zero.
>
> However, it is important to not that this is not a failure of Decision tree learning itself, and only that of greedy methods. If all the nodes are indeed jointly optimised, the optimal tree would indeed match the true labelling function.
>
>
> PS: ZanDT is a recent oblique tree construction method by Zantedeschi et al. (2020).

---

> > ### Comment · Reviewer_cuTu · 2023-12-04
> >
> > Thanks for your response. I still feel that the statement "Failure of Greedy Methods" needs to be explained in detail. In your provided example, I'm not convinced about the failure of a greedy methods, i.e., why info-gain=0 in that specific example? Even if it is true and root won't choose qi.x > 0, given that each sibling has opposite labels, it is still possible to construct a perfect classifier...
> >
> > Also, I agree with reviewer @67Vr regarding advantageous of the proposed approach on synthetic data only.
> >
> > I appreciate your comment regarding multi-class extension. However, I don't feel paper complete without proper evaluation on such tasks.
> >
> > For these reasons, I will maintain my original score of a reject.

---

### Author Response · Authors · 2023-11-21
**Addressing common points of concern**

We thank the reviewers for the comments. Here is a reply to the most common concerns of the reviewers.

Primary viewpoint:

The reviewers have viewed this paper as being one more way to construct a decision tree, and it just so happens to use a neural network. While that is a perfectly valid viewpoint, we feel the main value of the paper lies in Table 2 and its implication discussed in the last paragraph of Section 3. Elaborating it a bit more here: Feature learning is the "Dark Matter" of Neural networks. It is evident that it exists in the way Neural nets outperform kernel methods. But it is not clear what exactly it is, and when it happens. Here we claim that (at least in some restricted setting) feature learning can be viewed as seeking out surfaces of discontinuity in the label function. The DLGN architecture used in the model, and the ODT labelling function used simply make testing this hypothesis a possibility. This would not be possible (say) with the labelling function used by the CIFAR dataset or (say) a ReLU network. That we can use these discovered features in a trained network and use them in a completely different architecture like a decision tree, without doing any fine tuning, simply emphasises the usefulness of the features. It is analogous to using a the last layer features of a trained ResNet, as features in a linear classifier. The main difference is that these hyperplane features are fully interpretable.  We will emphasise this viewpoint in the revised submission.

Computational Concerns:

The computation effort into creating a DLGN DT is not significantly more than training a single model. For example if we assume the training time scales linearly with the dataset size, training a DLGN DT requires a factor of "depth of tree" more computation than training a DLGN. This is because, even though the number of DLGNs trained is equal to the number of nodes in the tree, the models other than the root are trained with much lesser data. See Figure 3. Also, the clustering is not computationally demanding, as it only requires to cluster "Number of neurons" vectors. This does not depend on the training data size and only depends on the depth and width of the DLGNs used for training.

Comparisons:

We trained a few more decision tree models (particularly Random forests and SDTs) and find that they also do comparably on UCI datasets, but perform noticeably worse on the synthetic datasets, particularly the high-dimensional data.

Binary classification:

The idea can be easily extended to multiclass methods as follows. The DLGN is a neural net architecture and can be trained on multiclass problems using cross-entropy in the same way as ReLU networks. Even multiclass-DLGNs still have neurons which correspond to half-spaces/hyperplanes. These can be used in DLGN-DT in the same way. The definition of discontinuity in the discontinuous hyperplanes does need to be changed appropriately for this.

Practical relevance to Decision Tree Learning:

The advantage of using DLGNs (and DLGN-DT) for learning decision trees only is visible in synthetic data, which is purposely designed to make greedy procedures fail. Natural data is not that adversarial, and hence the reason for immediately using DLGNs for learning decision trees is weak. We do agree on this front. But, we feel that this approach of analysing decision trees and the easy hyperplane viewpoint of a novel DLGN architecture add sufficient value, and open interesting potential directions for novel architectures and analyses.

---

### Meta-Review · Area_Chair_XnGC · 2023-12-08

**Metareview:**

All 3 reviewers recommend reject, based on a number of reasons. Please refer to the reviews. The experiments in particular need serious improvement. One important issue is that, however good the "features" or hyperplanes learned in nodes close to the root, the approach remains greedy, just as in traditional algorithms such as CART. This generally results in highly suboptimal, overly large trees. As noted in several reviews, a baseline to beat is the Tree Alternating Optimization (TAO) algorithm, which does optimize jointly over all the hyperplanes in a scalable way. Without comparing with that the claim in the abstract "The proposed algorithm outperforms all other decision tree construction procedures" doesn't hold.

That said, I (the AC) thinks that the paper's goal of understanding connections between feature learning in neural nets with other architectures, in particular decision trees is very interesting, and I look forward to an improved version of the paper.

**Justification For Why Not Higher Score:**

The paper needs considerable improvement.

**Justification For Why Not Lower Score:**

N/A

---

### Decision · Program_Chairs · 2024-01-16

Reject